# Hepatopancreas Proteomic Analysis Reveals Key Proteins and Pathways in Regulatory of Ovary Maturation of *Macrobrachium nipponense*

**DOI:** 10.3390/ani13060977

**Published:** 2023-03-08

**Authors:** Sufei Jiang, Hui Qiao, Hongtuo Fu, Zemao Gu

**Affiliations:** 1College of Fisheries, Shuangshui Shuanglü Institute, Huazhong Agricultural University, Wuhan 430070, China; 2Hubei Hongshan Laboratory, Wuhan 430070, China; 3Key Laboratory of Freshwater Fisheries and Germplasm Resources Utilization, Ministry of Agriculture and Rural Affairs, Freshwater Fisheries Research Center, Chinese Academy of Fishery Sciences, Wuxi 214081, China

**Keywords:** *Macrobrachium nipponense*, comparative proteomics, hepatopancreas, ovary maturation

## Abstract

**Simple Summary:**

The crustacean hepatopancreas was reported to play important roles in ovarian maturation, providing not only energy but also essential fatty acids and cholesterol required for the synthesis of steroid hormones. In addition, it is also an important site of vitellogenin synthesis in crustaceans. The role of crustacean hepatopancreas in regulating ovarian maturation in crustaceans has attracted more attention. However, the molecular mechanisms about the regulation are little known in scientific research. Our data indicated the key proteins of carbohydrate metabolism, lipid metabolism, amino acid metabolism, and lysosome pathways played important roles in hepatopancreas, as the ovaries developed to maturation in *Macrobrachium nipponense*. The results provide new insight into regulatory mechanisms of hepatopancreas in crustacean reproduction.

**Abstract:**

A TMT-based (Tandem Mass Tag) liquid chromatography-tandem mass spectrometry (LC-MS/MS) proteomics approach was employed to explore differentially expressed proteins (DEPs) and KEGG pathways in hepatopancreas of 5 ovary stages. In total, 17,999 peptides were detected, among which 3395 proteins were identified. Further analysis revealed 26, 24, 37, and 308 DEPs in HE-I versus HE-II, HE-II versus HE-Ⅲ, HE-Ⅲ versus HE-Ⅳ, and HE-Ⅳ versus HE-Ⅴ, respectively (HE-I, HE-II, HE-III, HE-IV, and HE-V means hepatopancreas sampled from ovary stage I to V.). Gene ontology (GO) analysis indicated that DEPs were significantly enriched in “catalytic activity”, “metabolic process”, and “cell” of 4 comparison groups in turn. Kyoto Encyclopedia of Genes and Genomes (KEGG) enrichment results showed that in hepatopancreas, as the ovaries developed to maturation, carbohydrate metabolism, lipid metabolism, amino acid metabolism, and lysosome played important roles in turn. The mRNA expression of 15 selected DEPs were consistent with proteome results by qPCR analysis. Further mRNA expression investigation results suggested 4 proteins (fatty acid-binding protein, NPC intracellular cholesterol transporter 1, Serine hydroxymethyltransferase, and Crustapin) were involved in ovary maturation. These results enhance the understanding of the regulatory role of hepatopancreas in *M. nipponense* ovary maturation and provide new insights for understanding the crustacean regulation mechanisms.

## 1. Introduction

During vitellogenesis in crustaceans, the ovary accumulates many nutrients, such as proteins and lipids, which are important energy and structural substances with vital roles in the reproductive process, biochemistry, and substance metabolism [1,2,3,4]. Hepatopancreas is the center of nutrients storage and metabolism in decapod crustaceans, and has an important role in ovarian lipid accumulation during ovarian development [5,6,7,8]. During ovary maturation, the hepatopancreas continuously produces vitellogenin and transports it via hemolymph to the ovary [9,10]. Once the yolk has reached a certain size, there is a significant decrease in the content of protein and lipids related to the corresponding increase of yolk in the hepatopancreas [11,12,13,14,15]. Although much is known about the role of the hepatopancreas in regulating ovarian maturation in crustaceans, the molecular mechanisms involved are less well understood.

*Macrobrachium nipponense*, also known as the oriental river prawn, is a commercial species mainly distributed in most freshwater areas of China, except Tibet and Qinghai [16]. It is very popular in the eastern China and the Yangtze River basin because of its high nutritional value and delicious taste. Pond-aquaculture of *M. nipponense* are prevalent with production exceeding 240,000 tons in China in 2021 [17]. Adult female *M. nipponense* have a short sexual maturity cycle [18]. During each breeding season, gonad maturation and larval development of *M. nipponense* are accelerated by increases in water temperature. This problem is particularly acute in female’s larva. Newborn female prawns develop to sexual maturity and lay eggs within 45 days after hatching by early autumn (known as “autumn reproduction”) [19]. Such rapid sexual maturation results in multiple generations and reduces resilience and survival rates [18]. A large amount of energy is used for such reproduction, resulting in significant decreases in the number of large-sized prawn, especially the female, which seriously affected the prawn aquaculture [20]. Therefore, it is urgent to carry out the mechanisms analysis of rapid ovary maturation providing insights to help address these problems.

Our previous studies of five ovary developmental stages detected many ovary maturations related genes with more dominant expression in the hepatopancreas compared within ovaries, which provided strong evidence for regulating role of the hepatopancreas in ovarian maturation [18,19,20,21]. Furthermore, comparative hepatopancreas transcriptomes of five ovary developmental stages were established [22] to provide a new field for studying fast ovary maturation in *M. nipponense*. In this study, a TMT-based (Tandem Mass Tag) liquid chromatography-tandem mass spectrometry (LC-MS/MS) proteomics approach was employed to explore differentially expressed proteins (DEPs) in hepatopancreas of 5 ovary stages. The study aimed to detect potential functional proteins and signaling pathways in *M. nipponense* hepatopancreas involved in regulating ovarian maturation. The results will provide new insight into the regulatory mechanisms involved in the role of the hepatopancreas in crustacean reproduction.

## 2. Materials and Methods

### 2.1. Experimental Samples Preparation

Adult female *M. nipponense* at 5 ovary stages (each stage contained 30 individuals, body weight (BW) ± SD: 2.08 ± 0.35 g) were sampled from Freshwater Fisheries Research Center Dapu Scientific Experimental Base (Wuxi, China). The different stages of ovarian development were determined based on color according to previous study [23]. The detail information of 5 ovarian development stages was listed in Table 1 and the histological observation was showed in Figure 1. For the experimental samples, each sample contained six prawns of same ovarian maturation stage with three biological repetitions (18 individuals per stage). The *M. nipponense* individuals selected were dissected after anesthesia with methane sulfonate (MS222). The hepatopancreas tissues for each of the five ovary stages were defined as HE-I, HE-II, HE-III, HE-IV, and HE-V respectively. Each hepatopancreas was dissected on ice and then stored in liquid nitrogen and then was stored at −80 °C until use.

### 2.2. Total Protein Extraction, Digestion and TMT Labeling

Lysis buffer was prepared as follows: 1 mL lysis buffer (40 mM Tris-HCl and 8 M urea) containing 1 μL protease inhibitor (Thermo Fisher Scientific, Shanghai, China), 5 μL phosphatase inhibitor, and 10 μL phenylmethylsulfonyl fluoride (PMSF) (Thermo Fisher Scientific, Shanghai, China). Once prepared, the buffer was placed on ice for a few minutes before use. Frozen hepatopancreas (100 μg) was lysed with ice-cold lysis buffer. The mixture was centrifuged at 12,000× *g* at 4 °C for 20 min after sonicated for 5 min. The supernatant was put in a pre-cooled tube, placed on ice for 10 min, and was violently shaken 2–3 times. The homogenate was then centrifuged at 15,000× *g* for 10 min at 4 °C, and the total protein was placed in a new pre-cooled tube. Finally, the protein concentration was tested using the BCA Protein Assay Kit (Beyotime, Shanghai, China). The quality of extracted proteins was tested by 12% SDS-PAGE (sodium dodecyl sulfate-polyacrylamide gel electrophoresis). Then 100 μg protein per sample were digested using trypsin (Promega, Madison, WI, USA) (trypsin ratio = 50:1) at 37 °C for 12 h and TMT reagents from TMT^®^ Mass Tagging Kits (Thermo Fisher Scientific, Karlsruhe, Germany) were used for peptide labeling, following the manufacturer’s instructions. 

### 2.3. Fraction Separation and LC-MS/MS Analysis

The labeled peptide was mixed and centrifuged at 20,000× *g* for 5 min. After that, conventional HPLC under alkaline conditions (Waters XBridge Shield C18 RP column, 3.5 µm, 4.6 × 250 mm, Shimadzu LC20AD) was used for the reverse gradient separation. About 60 fractions of the eluted peptide were combined in 12 fractions. They were drained using a vacuum concentrator according to the peak range shown by UV light.

Each sample was dissolved in solvent A/B (A: 0.1% formic acid in water; B: 80% acetonitrile with 0.1% formic acid) and centrifuged for 2 min at 20,000× *g*. The EASY-nLC™ 1200 UHPLC system (Thermo Fisher Scientific, Karlsruhe, BW, Germany) was used for sample separation. The flow rate was 250 µL/min with solvent B and the gradient was 0–6 min, 2–10%; 6–51 min, 10–20%; 51–58 min, 20–80%; 58–62 min, 80%; 62–63 min, 80–2%; 63–70 min, 2%. The peptides were then subjected to tandem mass spectrometry (MS) using a Q-Exactive™ HF-X mass spectrometer (Thermo Fisher Scientific, Karlsruhe, BW, Germany). The scanning range of the mass spectrometer was 350–1800 *m*/*z*. 

### 2.4. Data Processing and Bioinformatics Analysis

The original data file was imported into Proteome Discovery 2.2 software (Thermo Fisher Scientific, Karlsruhe, BW, Germany) for database retrieval and quantification of spectral peptides and proteins. BLASTALL (V2.2.26, E-value ≤ 10^−5^) was used for protein annotation against a reference genome and the RNA-sequencing results from the *M. nipponense* database and protein function analyses (NCBI Accession No. ASM1510439v1, BioProjects: PRJNA646023; accession No. SAMN27687877–SAMN27687891, Bioproject PRJNA830321) [22,24]. The proteins whose quantitation were significantly different in the HE-I versus HE-II, HE-II versus HE-III, HE-III versus HE-IV, and HE-IV versus HE-V groups (*p* < 0.05 and FC < 0.83 or fold change FC > 1.2) were defined as DEPs (differentially expressed proteins). The fold change of each protein was calculate by using average protein levels and *p* value was calculated using *t*-tests and corrected by the false discovery rate (*p* < 0.05).The principal-component analysis (PCAs) were performed using gmodels R (https://cran.r-project.org/web/packages/gmodels/index.html, accessed on 8 October 2021). Identified DEPs were annotated using the GO database (Gene Ontology) (http://geneontology.org/, accessed on 8 October 2021), Swissprot (http://www.uniprot.org/, accessed on 8 October 2021), COG (Clusters of Orthologous Groups) (https://www.ncbi.nlm.nih.gov/COG/, accessed on 8 October 2021), and KEGG pathways (www.kegg.jp, accessed on 8 October 2021) (E-value ≤ 10^−5^). DEPs were also subjected to functional GO terms and KEGG pathway enrichment analysis.

### 2.5. Real-Time PCR Validation and Statistical Analysis

Fifteen DEPs identified in this study were randomly chose to validate the proteomics analysis results by qPCR. The eukaryotic translation initiation factor 5A gene (EIF) was used as a reference gene [25]. Furthermore, the mRNA expression profiles and tissue distribution of four proteins across the five ovarian maturation stages were also analyzed. Tissues (H: heart, G: gill, HE: hepatopancreas, O: ovary, M: muscle) were used for total RNA extraction. The qPCR methods were followed by our previous study [26]. The relative expression levels were analyzed using the 2^−ΔΔCT^ [27] and the amplification efficiency of all primers was tested by establishing standard curve before the qPCR validation was performed (primers with amplification efficiency above 90% were used for the further step of qPCR). Primers were in Appendix A. The quantitative data were described using mean ± standard deviation. Statistical analyses were performed using SPSS 23.0 and one-way ANOVA was used to analyze statistical differences.

## 3. Results

### 3.1. Proteins Identification and Analysis 

In total, 17,999 peptides were detected in the *M. nipponense* hepatopancreas proteome. Most peptides were 7 and 17 amino acids long (Figure 2A). Protein mass results indicated that most proteins were either >100 kDa or 10–60 kDa (Figure 2B). In terms of the peptide number, most of the unique peptides contained fewer than 11. The protein’s sequence coverage results showed that 41.3% proteins had >10% sequence coverage and 23.74% proteins showed >20% sequence coverage (Figure 2D). PCA results (Figure 2E) showed the principal components of HE-I, HE-II, HE-III, and HE-IV groups were relatively concentrated, suggesting that there was no significant difference in the content and types of proteins identified among the four groups. HE-V group showed significant differences from the other four groups.

### 3.2. Comparison of Proteome Profiles and DEP Discovered

Comparison of the proteome profiles in HE-I versus HE-II, HE-II versus HE-III, HE-III versus HE-IV, and HE-IV versus HE-V were performed to identify DEPs involved in ovarian maturation (*p* < 0.05 and FC < 0.83 or fold change FC > 1.2) with the 3395 proteins found. Further analysis revealed 26, 24, 37, and 308 DEPs in HE-I versus HE-II, HE-II versus HE-III, HE-III versus HE-IV, and HE-IV versus HE-V, respectively. The HE-IV versus HE-V group had 308 DEPs, which were significantly much more than other three groups. The details of up- or downregulated information of all DEPs in different groups were listed in Figure 3A. Identified DEPs were functionally annotated to the COG, GO, Swissprot, and KEGG database and the results were displayed in Figure 3B. The volcano plots of each group of DEPs are shown in Figure 4A–D. The top three most up/downregulated DEPs in the four comparison groups were listed in Table 2.

### 3.3. GO Enrichment

GO enrichment was performed to functionally identify the DEPs *(p* < 0.05). All DEPs were attributed to biological processes (BP), cellular components (CC) and molecular functions (MF). In HE-I versus HE-II, there were 11, 4, and 13 GO terms in CC, MF, and BP, respectively, “catalytic activity” had the most classification terms (14 DEPs) (Figure 5A). There were 12, 4, and 13 GO terms in the three groups and “metabolic process” was significantly enriched (10 DEPs) in the HE-II versus HE-III (Figure 5B). In HE-III versus HE-IV comparison the three groups contained 10, 5, and 14 GO terms, respectively, and “cell” had the most classification terms (21 DEPs) (Figure 5C). There were 14, 8, and 23 enriched terms in CC, MF, and BP, respectively, in HE-IV versus HE-V group, and “cell” terms were the highest classification terms (149 DEPs) (Figure 5D).

### 3.4. KEGG Enrichment

KEGG pathway enrichment was performed for all DEPs (Table 3). There were seven enriched pathways in the HE-I versus HE-II group, and 85% of which were related to carbohydrate metabolism (Figure 6A). “Various types of N-glycan biosynthesis” and “N- glycan biosynthesis” were both significantly enriched (*p* < 0.05). There were 11 enriched pathways in the HE-II versus HE-III group and 9 of the 11 KEGG pathways were related to carbohydrate, amino acid, and lipid metabolism (Figure 6B). Among these, “Fatty acid degradation” was the only significantly enriched pathway (*p* < 0.05). The HE-III versus HE-IV group had 17 enriched pathways and 9 of which were involved in amino acid and energy metabolism (Figure 6C). The other 8 mainly focused on a series of oxidation-reduction reactions and transport such as “Peroxisome” and “Endocytosis”. “Glycine, serine, and threonine metabolism” was the only significantly enriched pathway (*p* < 0.05). The HE-IV versus HE-V comparison had the most enriched pathways (67), 40 of which were involved in amino acid, carbohydrate, and lipid metabolism (Figure 6D). “Arginine biosynthesis s” and “Lysosome” were significantly enriched (*p* < 0.05).

### 3.5. DEP Validation

Fifteen DEPs with large expression quantitation fold change in four different comparison groups (HE-I vs. HE-II, HE-II vs. HE-III, HE-III vs. HE-IV, HE-IV vs. HE-V) were selected from Table 2 and Table 3 for qPCR validation. They were: arylsulfatase, alpha-(1,6)-fucosyltransferase, fatty acid-binding protein, NPC intracellular cholesterol transporter 1), serine hydroxymethyltransferase, choline O-acetyltransferase, sarcosine oxidase, sodium-dependent multivitamin transporter, protein 1gg-1, glutamine synthetase, glyceraldehyde-3-phosphate dehydrogenase, vitellogenin, cystatin-1, crustapin, legumain-like protein. The qPCR results were consistent with the data from proteomics (Figure 7) (*p* < 0.05).

### 3.6. Key Proteins Involved in Ovary Maturation

We also analyzed the hepatopancreas and ovary transcription levels of the genes encoding 4 proteins (*FABP*, *Nct-1*, *Shf* and *Crp*) in each of the five ovarian developmental stages (Figure 8), which were randomly selected from 15 validated candidate key proteins, one in each comparison groups. The mRNA expression results showed that there were two distinct upregulated expressions of *FABP* in hepatopancreas and its expression of ovary was upregulated significantly during ovarian development (Figure 8A). However, *Nct*-1 expressions differed in hepatopancreas and ovary (Figure 8B). In hepatopancreas, its expression was positively correlated and increased significantly with ovarian maturation, whereas, in the ovary, its expression was negatively correlated with ovarian development. *Shf* and *Crp* had similar expression patterns in the ovary and hepatopancreas (Figure 8C,D). The highest expression of these two proteins in the ovary was observed during vitellogenesis (stage III), whereas their highest expressions in the hepatopancreas occurred in stage II.

We also examined the tissue transcript distribution of *FABP*, *Nct-1*, *Shf*, and *Crp* (Figure 9). All four proteins displayed extremely high expression levels in hepatopancreas (*p* < 0.05). mRNA expression of *FABP*, *Nct-1*, *Shf*, and *Crp* was higher in the ovary than in the heart, gill, and muscle (Figure 9A–D). The *FABP* and *Nct*-1 expression in the ovary was slightly higher than in other tissues, although not significantly so (*p* > 0.05) (Figure 8A,B). However, *Shf* and *Crp* expression in ovary was significantly higher than in other tissues (*p* < 0.05) (Figure 9C,D).

## 4. Discussion

The hepatopancreas in crustaceans is an important organ which plays a key multifunctional role in various physiological processes such as growth, immune response, metabolism, and reproduction [28]. The ovarian development process in crustaceans is very complex, including oogenesis, vitellogenesis, hormone secretion, and nutrient deposition [29,30]. During crustacean ovary development, the nutrients are transported to the ovaries from the hepatopancreas and the content of protein and lipid in hepatopancreas change regularly [31,32,33,34,35]. Hepatopancreas transcriptomes of 5 ovary stages were compared to gain new insights into the role of the hepatopancreas in ovarian maturation in *M. nipponense*. Changes in proteins identified in hepatopancreas will provide further information for functional investigations of the molecular regulatory mechanisms involved crustaceans’ sexual maturation [22]. In present study, a TMT-based proteomics analysis was used to identify key proteins and pathways during ovarian development of *M. nipponense*.

Ovarian development is a continuous, gradual process in crustaceans. Based on the current results, there were relatively few DEPs between the HE-I versus HE-II, HE-II versus HE-III, HE-III versus HE-IV. By contrast, the number of DEPs in the HE-IV versus HE-V group was ten times that of the other comparisons. That was because the ovary development underwent dramatic and complex physiological changes from maturation (stage IV) to emptying (stage V). The GO enrichment results also supported the continuous, gradual process of ovarian maturation. The results also showed that numerous DEPs were mainly enriched in biological processes (BP) and cellular components (CC) suggested that ovary development involved complex physiological and biochemical processes. 

Based on our understanding of transcriptomes and metabolomics related to ovarian development in *M. nipponense*, the five developmental stages were divided into two parts: oogenesis (IV to V and I to II) and vitellogenesis (II to III and III to IV) [21,36]. From the KEGG enrichment results, 85% of the KEGG pathways identified were related to carbohydrate metabolism in the HE-I versus HE-II group, among which the most significantly enriched pathways were “Various types of N-glycan biosynthesis” and “N-glycan biosynthesis”. The only enriched DEP was alpha-(1,6)-fucosyltransferase, which involved in the synthesis of active ribose, providing raw materials for the formation of precursor sugar nucleotides and participating in the synthesis and assembly of fucose [37]. Recent research has highlighted the involvement of fucose in reproductive performance in mice [38]. These findings further suggested that carbohydrate metabolism had a crucial part in oogenesis in *M. nipponense* and was perhaps major source of energy. The top most significantly up-regulated genes have also been reported to be involved in reproduction in other animals, arylsulfatase A type is essential for sperm binding to the zona pellucida in mice [39], vanin-2 protein has an activation role in bovine follicles [40], and fatty acid-binding protein has a promoting effect on ovarian maturation in crustaceans (*Portunus trituberculatus*) and bovines [41,42]. In the current study, *FABP* mRNA expression in ovary was significantly upregulated during ovarian development suggesting its potential promoting role in ovarian maturation.

Yolk and lipid production began at ovary stage II reflected by the gradual enlargement of the ovary and the color gradually turning yellow. During this progress, lipid and protein contents significantly increase in the ovary, which was related to the corresponding decrease of these substances in the hepatopancreas [43]. From the comparison of HE-II versus HE-III in this study, the KEGG enrichment results revealed that the “Fatty acid degradation” pathway has the most significant changes. This suggested that numerous lipids related to the formation of egg lipids were preserved in the hepatopancreas, which were decomposed and transferred to the developing eggs during ovarian development. Peroxisomal acyl-coenzyme A oxidase 1 was reported to be a rate-limiting enzyme in fatty acid β-oxidation which is involved in estrogen signaling in brown trout *Salmo trutta f. fario* [44,45,46]. Cholesterol is an important substrate for ovarian steroidogenesis, and the significantly upregulated gene *NPC* encodes intracellular cholesterol transporter 1 coded for a glycoprotein which processed low-density lipoprotein importing cholesterol [47,48]. In this study, the mRNA expression of Nct-1 was positively correlated with ovarian maturation in hepatopancreas while negatively in ovary, suggesting that it was involved in steroid hormone synthesis that inhibited ovarian maturation. Fibulin-1, which is a secreted protein associating with elastic matrix fibers, is induced by progesterone in human ovarian epithelial stromal cells [49,50]. Choline O-acetyltransferase shows an obvious decrease during in this period and its expression is regulated by estradiol in mammals [51,52]. This result also suggests that sex hormone levels change during this period. The results indicated that carbohydrate metabolism from stage II to III was significantly reduced, as indicated by the significant decrease in with the GDP-mannose 4,6 dehydratase.

From ovary stage III to IV in *M. nipponense*, the ovary expands rapidly and its color changes from light to dark green as yolk accumulation accelerates. In the HE-III versus HE-IV group, nine of these pathways were involved in amino acid and energy metabolism and other eight mainly focused on series of oxidation-reduction reactions. The “Glycine, serine, and threonine metabolism” pathway had the most significant changes, and sarcosine oxidase and serine hydroxymethyltransferase were the top-most upregulated proteins. Previous research showed that glucose, pyruvate, glutamine and glycine increased significantly during maturation of bovine oocytes [53]. However, the mechanisms involved in oocytes maturation required further investigation. The relative RNA expression of sarcosine oxidase increases in bovine oocytes after the preovulatory luteinizing hormone surge [54], whereas serine hydroxymethyltransferase is necessary for ovarian cancer tumor growth and cell migration in mice [55]. In the current study, serine hydroxymethyltransferase showed the highest mRNA expression during ovary stage Ⅲ, suggesting it had an important role in vitellogenesis. The detailed roles of these proteins in terms of oxidation-reduction reactions in ovarian maturation remain to be determined in further researches.

From stage IV to V in *M. nipponense*, the ovary undergoes dramatic changes, reflecting the change from being full of mature eggs to being completely empty. The comparison between these two stages also confirmed this phenomenon as indicated by the numerous DEPs and signaling pathways. Significant decreases in several proteins (vitellogenin, NPC intracellular cholesterol transporter 1, and glyceraldehyde-3-phosphate dehydrogenase) were also demonstrated, while “Arginine biosynthesis” and “Lysosome” were significantly enriched. Arginine is a versatile amino acids with important roles in protein synthesis, as well as serving as a precursor for compounds such as nitric oxide, polyamines, and agmatine involved in sexual reproduction, hormone metabolism, and so on [56,57,58]. Aspartate aminotransferase controls enzyme activity during the spawning phase in fish central metabolism [59]. Argininosuccinate synthase and argininosuccinate lyase were important in nitric oxide synthesis which has a wide range of reproductive functions [60], such as in gonadotrophin secretion, estradiol synthesis, follicle survival, and ovulation [61]. The lysosome contains more than 60 hydrolytic enzymes involved in its digestive function and has important roles in endocytosis, exocytosis, and autophagy. Recent research highlighted new roles of lysosomes in mammalian females. In the ovary, lysosomes have important roles in steroidogenesis synthesis by free cholesterol and regulation of follicular atresia, follicle rupture during ovulation, and luteal regression [62]. In previous works, three genes (encoding legumain-like protein, crustapin, and cathepsin L) from “Lysosome” pathways were confirmed to promote ovarian maturation by RNA interference technology in *M. nipponense* [18,20,63]. Protein lgg-1 was one of important members in autophagy. Autophagy played essential roles in cell fate decision and cellular homeostasis maintenance, as well as reproduction. New reports indicated that autophagy process blockage may lead to abnormal reproduction [64]. Therefore, more efforts were needed to elucidate the mechanism of lysosome and autophagy enzymes in crustacean female reproduction. In this study, a large number of DEPs were screened, but their roles in reproduction have not been clarified. Moreover, an important direction of future research is to demonstrate the novel functions of these proteins in reproduction in crustaceans.

## 5. Conclusions

The present study, which applied TMT-based proteomics approach in hepatopancreas during female *M. nipponense* ovarian maturation, provides an overview of regulatory mechanisms of hepatopancreas in crustacean reproduction. The DEPs data in HE-I versus HE-II, HE-II versus HE-Ⅲ, HE-Ⅲ versus HE-Ⅳ, and HE-Ⅳ versus HE-Ⅴ, respectively, indicating the HE-V has the most different proteins than other four stages. KEGG enrichment results showed that in hepatopancreas, as the ovaries developed to maturation, carbohydrate metabolism, lipid metabolism, amino acid metabolism, and lysosome played important roles. Table 2 and Table 3 conclude the candidate proteins from this study based on the DEPs and KEGG enrichment analysis. qPCR analysis proves the proteome results was consistent with mRNA expression results. Further investigation of 4 randomly selected candidate proteins (fatty acid-binding protein, NPC intracellular cholesterol transporter 1, serine hydroxymethyltransferase, and Crustapin) showed their involvement in ovary maturation. So far, more efforts have been made in screening key pathways, proteins and genes related to ovary maturation of *M. nipponense* and nutrient metabolism-related pathways and proteins are becoming increasingly prominent. For further study, we will aim to analyze the spatial and temporal expression patterns and biological functions of these candidate proteins and their regulation relationship in ovarian maturation.

## Figures and Tables

**Figure 1 animals-13-00977-f001:**
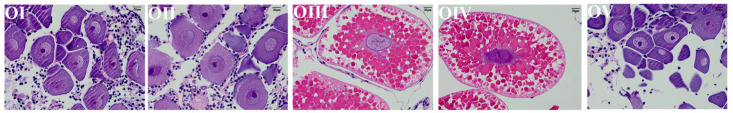
Histological observation of 5 ovarian stages.

**Figure 2 animals-13-00977-f002:**
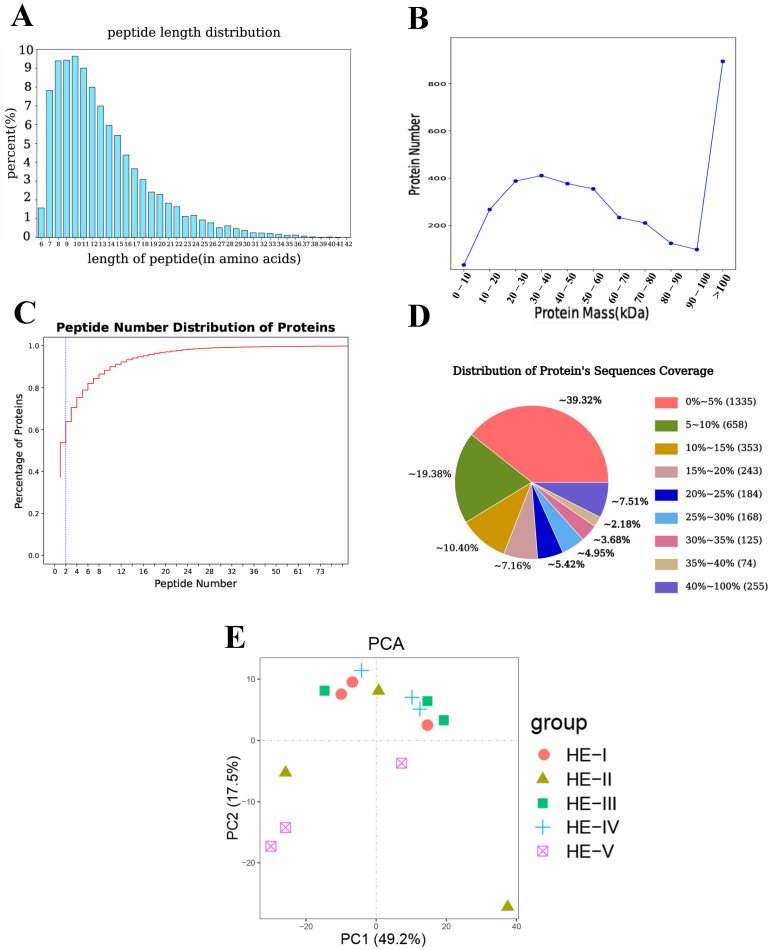
Summary information of hepatopancreas proteomes during ovary maturation of *Macrobrachium nipponense.* (**A**) The peptide length (amino acids). (**B**) Protein mass. (**C**) Spectrum percent of peptides. (**D**) Distribution of protein’s sequences coverage. (**E**) Proteins separation of PCA.

**Figure 3 animals-13-00977-f003:**
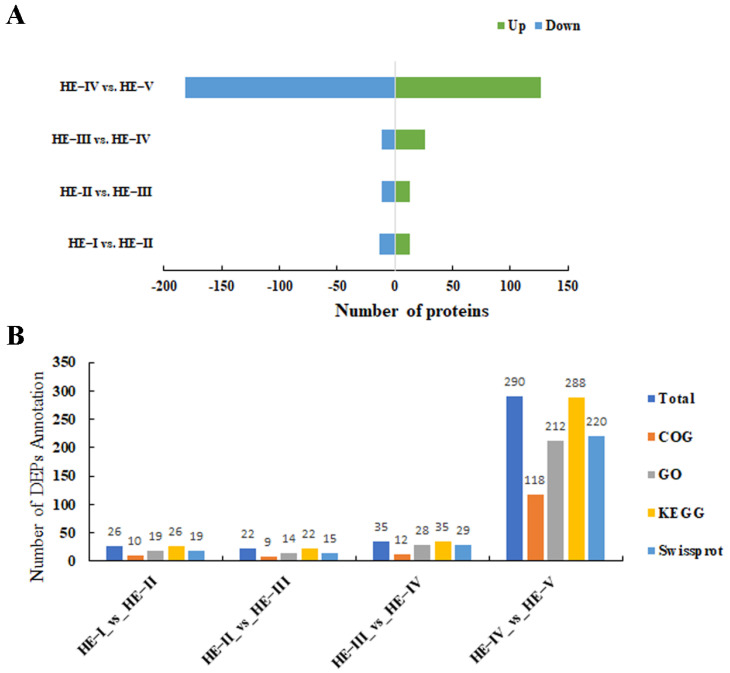
The analysis of (**A**) number of up- and downregulated DEPs. (**B**) Functional classification of DEPs with the COG, GO, Swissprot, and KEGG database in HE-I vs. HE-II, HE-II vs. HE-III, HE-III vs. HE-IV, and HE-IV vs. HE-V groups.

**Figure 4 animals-13-00977-f004:**
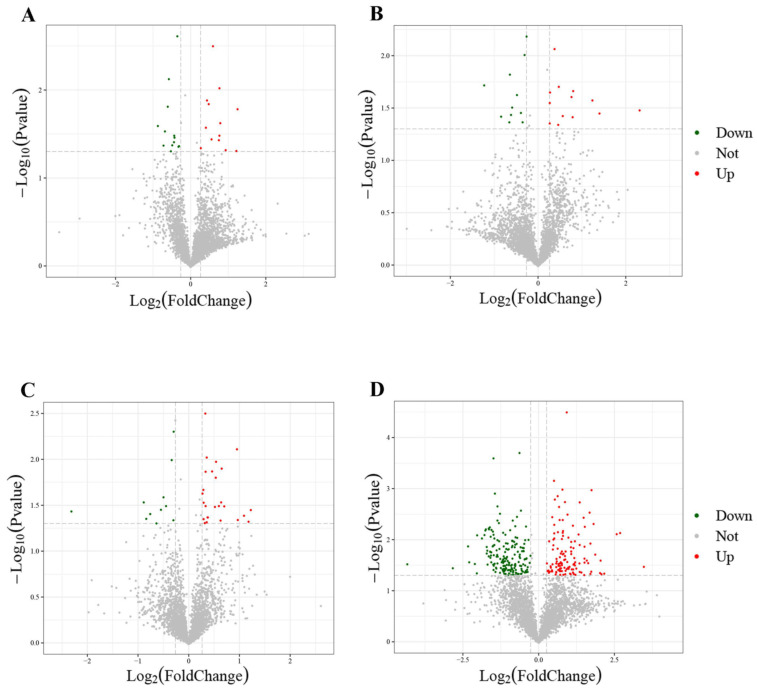
Volcano plot diagram of DEPs in different comparison groups. (**A**) HE-I vs. HE-II, (**B**) HE-II vs. HE-III, (**C**) HE-III vs. HE-IV, (**D**) HE-IV vs. HE-V. Red dot indicates upregulated DEPs and green dot represents downregulated DEPs.

**Figure 5 animals-13-00977-f005:**
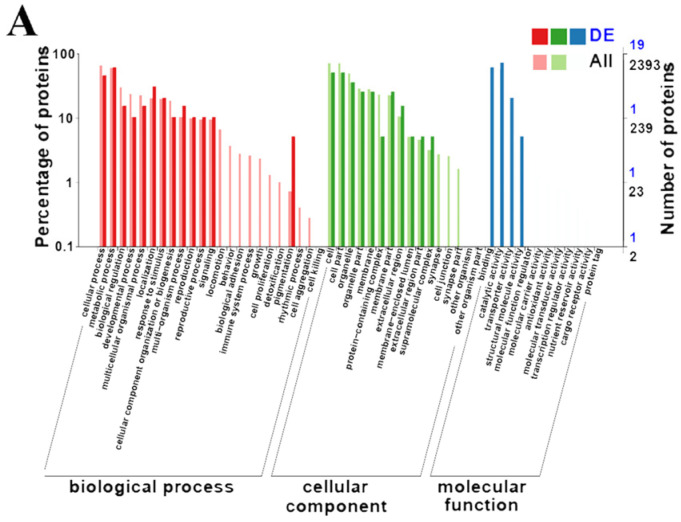
GO enrichment analysis of DEPs in different comparison groups. (**A**) HE-I vs. HE-II, (**B**) HE-II vs. HE-III, (**C**) HE-III vs. HE-IV, (**D**) HE-IV vs. HE-V.

**Figure 6 animals-13-00977-f006:**
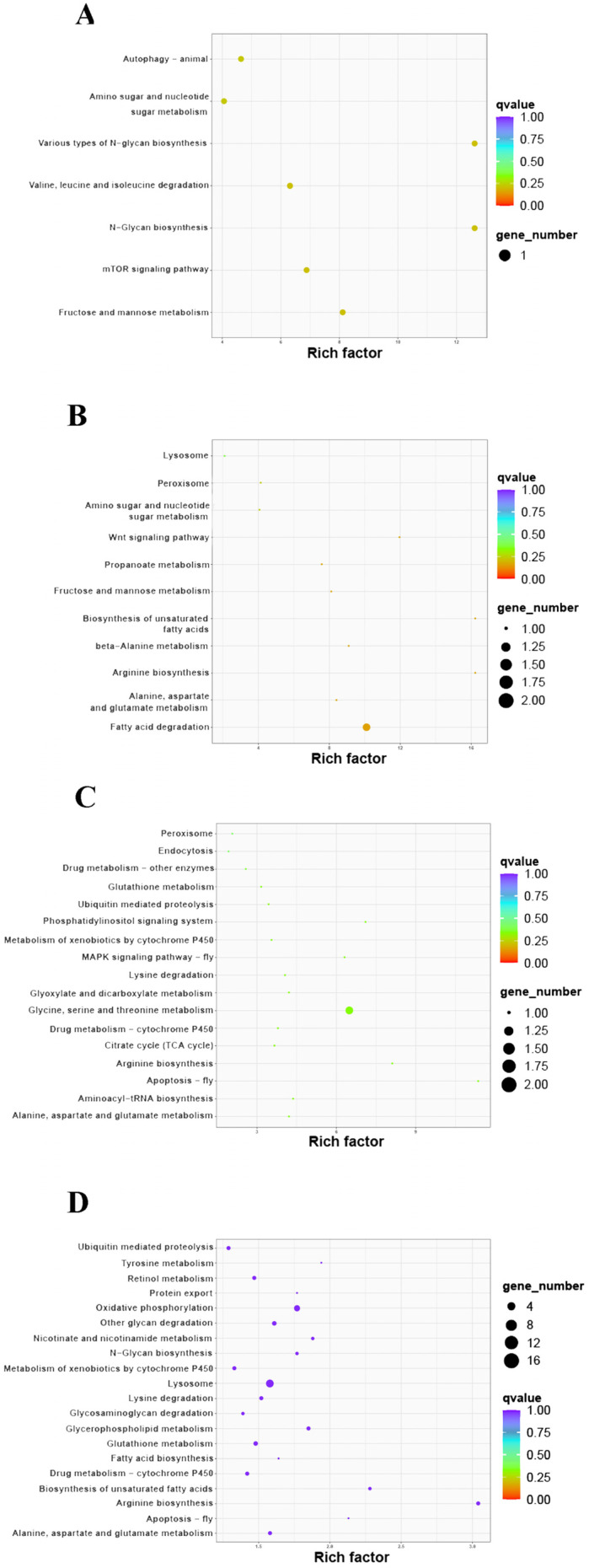
KEGG pathway enrichment analysis of DEPs in different comparison groups. (**A**) HE-I vs. HE-II, (**B**) HE-II vs. HE-III, (**C**) HE-III vs. HE-IV, (**D**) HE-IV vs. HE-V.

**Figure 7 animals-13-00977-f007:**
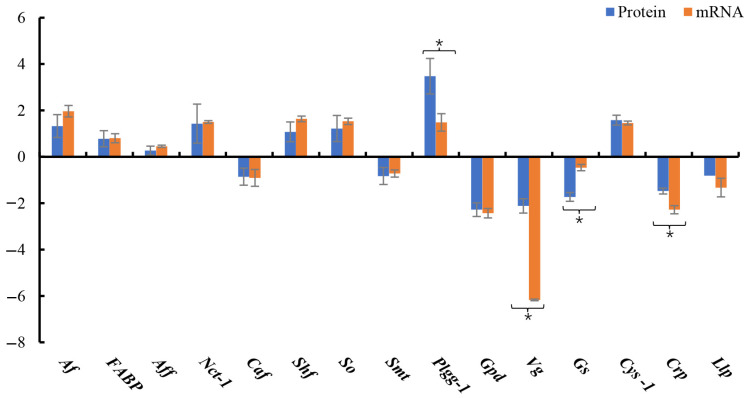
qPCR validation. Comparison of the selected genes by Protein-Seq and qPCR. The fold changes were calculated by 2^−ΔΔCT^. Data are shown as means ± SD (*n* = 3). * means significant differences (*p* < 0.05).

**Figure 8 animals-13-00977-f008:**
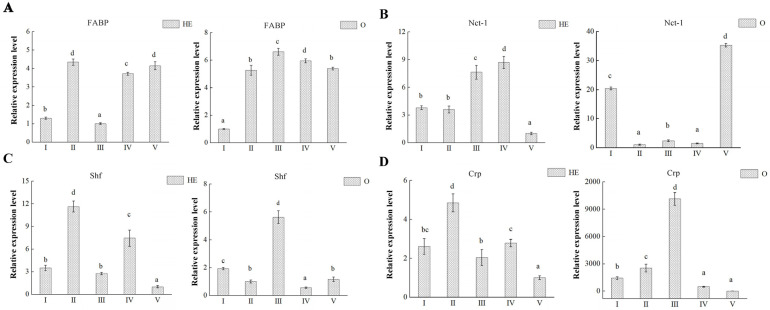
mRNA expression profiles of 4 proteins in both hepatopancreas and ovary during five ovary stages. (**A**) *FABP*, (**B**) *Nct-1*, (**C**) *Shf*, (**D**) *Crp*. Data are shown as mean ± SD (*n* = 3). Statistical analyses were performed with one-way ANOVA analysis. a, b, c, and d denote significant differences (*p* < 0.05) and same letter means no significant difference (*p* > 0.05).

**Figure 9 animals-13-00977-f009:**
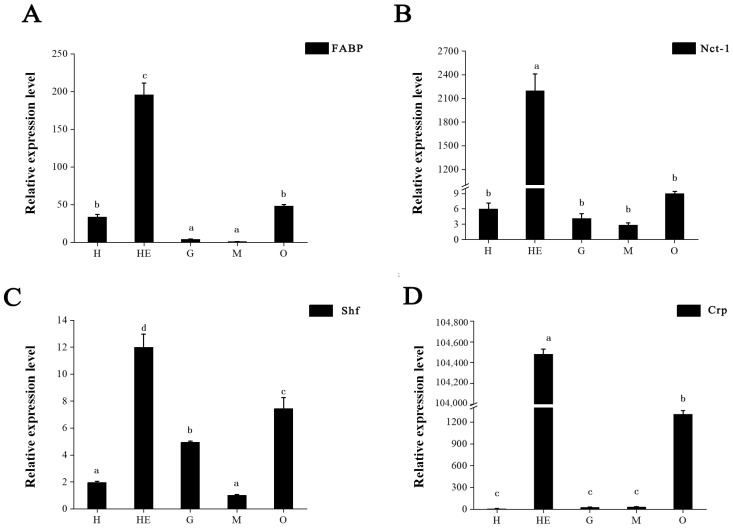
mRNA expression profiles of 4 proteins in different tissues. (**A**) *FABP*, (**B**) *Nct-1*, (**C**) *Shf*, (**D**) *Crp*. H: heart, G: gill, HE: hepatopancreas, O: ovary, M: muscle. Data are shown as mean ± SD (*n* = 3). Statistical analyses were performed with one-way ANOVA analysis. Lowercase letters on the columns (a, b, c, and d) denote significant differences (*p* < 0.05) and same letters mean no significant difference (*p* > 0.05).

**Table 1 animals-13-00977-t001:** Characteristics of 5 ovarian development stages.

Ovary Stages	The Description of Characteristics
OI	transparent, undeveloped stage
OII	yellow, developing stage
OIII	light green, nearly-ripe stage
OIV	dark green, ripe stage
OV	gray, spent stage

**Table 2 animals-13-00977-t002:** Top 3 up/down regulated DEPs identified in hepatopancreas of *M. nipponense* in four compared groups.

Comparison Group	Description	Regulation	Fold Change
HE-I vs. HE-II	Arylsulfatase	up	2.37
	Vanin-like protein 2	up	2.32
	Fatty acid-binding protein	up	1.70
	sodium-dependent dicarboxylate transporter	down	0.54
	Protein Hook homolog	down	0.62
	Probable 3-hydroxyisobutyrate dehydrogenase	down	0.65
HE-II vs. HE-III	NPC intracellular cholesterol transporter 1	up	2.64
	Pteridine reductase 1	up	2.36
	Fibulin-1	up	1.74
	Choline O-acetyltransferase	down	0.55
	GDP-mannose 4,6 dehydratase	down	0.63
	Arylsulfatase	down	0.64
HE-III vs. HE-IV	Sarcosine oxidase	up	2.33
	Serine hydroxymethyltransferase	up	2.13
	Origin recognition complex subunit 1	up	1.95
	sodium-dependent multivitamin transporter	down	0.56
	Myosin-VIIa	down	0.59
	ABC transporter C family member 10	down	0.70
HE-IV vs. HE-V	Protein lgg-1	up	11.10
	Epsin	up	6.45
	Aromatic amino acid aminotransferase	up	5.59
	NPC intracellular cholesterol transporter 1	down	0.04
	Glyceraldehyde-3-phosphate dehydrogenase	down	0.14
	Vitellogenin	down	0.20

**Table 3 animals-13-00977-t003:** DEPs of KEGG enrichment pathways (*p* < 0.05).

Comparison Group	Pathways	Pathway ID	DEP Number	Description	Regulation	Fold Change
HE-I vs. HE-II	N-Glycan biosynthesis	map00510	1	Alpha-(1,6)-fucosyltransferase	up	1.21
	Various types of N-glycan biosynthesis	map00513	1	Alpha-(1,6)-fucosyltransferase	up	1.21
HE-II vs. HE-III	Fatty acid degradation	map00071	2	peroxisomal acyl-coenzyme A oxidase 1	up	1.20
				choline O-acetyltransferase	down	0.55
HE-III vs. HE-IV	Glycine, serine and threonine metabolism	map00260	2	sarcosine oxidase	up	2.33
				serine hydroxymethyltransferase	up	2.13
HE-IV vs. HE-V	Arginine biosynthesis	map00220	4	aspartate aminotransferase	up	1.49
				Glutamine synthetase	down	0.31
				Argininosuccinate synthase	down	0.54
				argininosuccinate lyase	down	0.74
	Lysosome	map04142	14	V-type proton ATPase subunit d 1	up	2.84
				Sphingomyelin phosphodiesterase	down	0.41
				ceroid-lipofuscinosis neuronal protein 7	up	1.31
				Cystatin-1	up	3.01
				NPC intracellular cholesterol transporter 1	down	0.04
				Crustapin	down	0.32
				AP-1 complex subunit mu-1-I	down	0.61
				Arylsulfatase	down	0.35
				AP-3 complex subunit delta	down	0.77
				Mite group 2 allergen Lep d 2	down	0.33
				Beta-galactosidase 1	down	0.45
				legumain-like protein	down	0.44
				Cathepsin L	down	0.52

## Data Availability

The mass spectrometry proteomics data have been deposited to the ProteomeXchange Consortium (http://proteomecentral.proteomexchange.org, accessed on 30 September 2022) via the PRIDE partner repository with the dataset identifier PXD037141.

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
