# Peer review of "Hepatopancreas Proteomic Analysis Reveals Key Proteins and Pathways in Regulatory of Ovary Maturation of Macrobrachium nipponense"

_animals, 2023, doi:10.3390/ani13060977_

Round 1

Reviewer 1 Report

In the manuscript “Hepatopancreas proteomic analysis reveals key proteins and pathways in regulatory of ovary maturation of Macrobrachium nipponense”, the author reported the analysis of proteome of hepatopancreas from female of 5 ovarian developmental stages, and identified several import proteins and pathways related to ovary development. Further analysis combined with transcriptome data uncovered 4 key proteins involved ovary maturation. The results of the present study provided important information for crustacean reproduction regulation and are very meaningful for aquaculture development. However, some concerns should be addressed for it to be suitable to publish.

1. The samples used for this study are from female individuals of 5 developmental stages. However, the stage of samples was only based on the observation of the ovarian appearance as described in the manuscript. Histological examinations are needed for rigorous purpose.

2. Line 102, the producer information of protease inhibitor, phosphatase inhibitor, phenylmethylsulfonyl fluoride and so on should be provided.

3. Line 107, what’s the meaning of homogenate was “removed”? maybe typo?

4. Line 136-137, should be “The proteins whose quantitation were …”.

5. Line 153, Table “S1” is appeared in the main manuscript? And also, the full name of each gene is needed when it appeared for the first time.

6. Line 149, why choose these 4 proteins for further transcriptional analysis?

7. The statistical analysis algorithms description is missing in the Method section.

8. Lines 173-175, Before investigating the different expression patterns of different samples/stages, the Principal Component Analysis (PCA) should be conducted, which could provide a general landscape of expression patterns.

9. Line 176, the sentence “The EDPs in … with 308 number” should be rewritten to make it more readable.

10. Figure 2, the results of (A) and (B) are paradoxical. Such as, the number of DEPs in HE-IV vs HE-V are roughly around 50 in (A), but the annotated DEPs of the same group are more than 100 in (B).

11. Figure 3, the results are also not consistent with Figure 2A.

12. Figure 6A is in a way a repeat of Table 2&3 and may be considered to delete. Statistical significance should be labelled in Figure 6B.

13. The significance label letters should be reorganized.

Author Response

Dear Reviewer,

Thank you very much for your comments and suggestions. The valuable comments from you not only helped us with the improvement of our manuscript, but suggested some ideas for future studies.

Below you will find our responses to your comments:

  1. The samples used for this study are from female individuals of 5 developmental stages. However, the stage of samples was only based on the observation of the ovarian appearance as described in the manuscript. Histological examinations are needed for rigorous purpose.

Response:Thanks for your kind suggestions and we add histological photos to match ovarian appearance. Please find new Figure 1 in revised manuscript.

  1. Line 102, the producer information of protease inhibitor, phosphatase inhibitor, phenylmethylsulfonyl fluoride and so on should be provided.

Response: We add the producer information of protease inhibitor, phosphatase inhibitor, phenylmethylsulfonyl fluoride in the revised manuscript revised manuscript (Thermo Fisher Scientific, Shanghai, China).

  1. Line 107, what’s the meaning of homogenate was “removed”? maybe typo?

Response: Thanks for your kindly reminding. Here we changed to “The supernatant of tissue homogenate was put to a pre-cooled tube and placed on ice for 10 min, and was violently shaken 2-3 times.”

  1. Line 136-137, should be “The proteins whose quantitation were …”.

Response: Thanks for your kindly reminding. Here we add “were”.

  1. Line 153, Table “S1” is appeared in the main manuscript? And also, the full name of each gene is needed when it appeared for the first time.

Response: Thanks for your kindly reminding. We delete the table S1 here, and we add the full name of each gene in table S1 when it appeared for the first time.

  1. Line 149, why choose these 4 proteins for further transcriptional analysis?

Response: For candidate key proteins screening, first we defined the differentially expressed proteins (DEPs) according to the expression quantitation fold change. Then the DEPs were also subjected to functional GO terms and KEGG pathways enrichment. The top 3 up/down regulated DEPs and the ones which were in GO and KEGG enrichment pathways with P<0.05, were the candidate key proteins. All these candidate key proteins were listed in Table 2 and Table 3. Fifteen DEPs with large expression quantitation fold change in four different comparison groups (HE-â…  vs. HE-â…¡, HE-â…¡ vs. HE-â…¢, HE-â…¢ vs. HE-â…£, HE-â…£ vs. HE-â…¤), were selected from Table 2 and Table 3 for qPCR validation. For further investigation, 4 proteins were selected randomly from 15 validated candidate key proteins, one in each comparison groups. We also corrected the description in the manuscript.

  1. The statistical analysis algorithms is missing in the Method section.

Response: We add the statistical analysis description in the Method section, 2.5 Real-time PCR validation and statistical analysis. “The quantitative data were described mean ± standard deviation. Statistical analyses were performed using SPSS 23.0 and One-way ANOVA was used to analyze statistical differences.”

  1. Lines 173-175, Before investigating the different expression patterns of different samples/stages, the Principal Component Analysis (PCA) should be conducted, which could provide a general landscape of expression patterns.

Response: Thanks for your comments and we add a principal component analysis (PCA) before investigating the different expression patterns of different stages. Please find in “3.1 Proteins identification and analysis” in revised manuscript.

  1. Line 176, the sentence “The EDPs in … with 308 number” should be rewritten to make it more readable.

Response:We rewrite this sentence. “The HE-â…£versus HE-â…¤group had 308 DEPs, which were significantly much more than other three groups.”

  1. Figure 2, the results of (A) and (B) are paradoxical. Such as, the number of DEPs in HE-IV vs HE-V are roughly around 50 in (A), but the annotated DEPs of the same group are more than 100 in (B).

Response: Thank you for your comments. Yes, we are sorry to make mistakes in Figure 2A when moving the abscissa name to the left. In the revised manuscript, we correct this mistake.

  1. Figure 3, the results are also not consistent with Figure 2A.

Response: Yes, the Figure 2A was mistake and we correct this figure in the revised manuscript. Thank you for your comments.

  1. Figure 6A is in a way a repeat of Table 2&3 and may be considered to delete. Statistical significance should be labelled in Figure 6B.

Response: Thanks for your comments. We delete the Figure 6A. We also correct the Figure 6B with statistical significance labelled.

  1. The significance label letters should be reorganized.

Response: The significance label letters in Figure 7 were confused, we made a new Figure with separated hepatopancreas and ovary expressions.

Reviewer 2 Report

The manuscript studies the characterization of the hepatopancreas proteome in the crustacean Macrobrachium nipponense and the roles of several proteins in ovary maturation.

 General comments

 Several figures are of low resolution and I could not read the text (Figures 1, 4, 5 and 6).

 I think that the following data should be included in the supplementary material in excel file format (these data help readers and reviewers to understand and analyze the results obtained):

-The complete lists of differentially expressed proteins DEPs (comparison group, description, fold change and p-value)

-The complete list of significantly enriched GO terms and KEGG pathways (description, p-values )

 The conclusions are too general. There is no information about the specific results achieved by the authors. I would suggest the authors to rewrite their conclusions.

 English should be revised because some sentences are not clear.

 Specific comments

 Line 29. Please, explain, in the abstract, the meaning of the acronyms HE-I, HE-II, HE-III, HE-IV and HE-V.

 Lines 37-38. It should read “were involved in ovary maturation” instead of “were involved ovary maturation”

 Lines 61-62. Please, revise this sentence: “M. nipponense has characteristic with short sexual maturation cycle”.

 Lines 119-120: It should read “About 60 fractions of the eluted peptides were combined in 12 fractions and drained by vacuum concentrator” instead of “About 60 fractions of the eluted peptide were combined 12 fractions and drained by vacuum concentrator”

 Line 122. FA: formic acid?

 Lines 122-123. Please, revise this sentence: “Each sample was dissolved in solvent A/B (A: 0.1% FA in water; 80% acetonitrile with 0.1% FA), and was centrifuged for 2 min at 20000 g”. Which is the solvent B?

 Lines 138-139. Has the P-value been adjusted for multiple testing correction (for example by the false discovery rate).

 Lines 151-152. The 2-ΔΔCT method used to analyze the mRNA expression levels is valid only if the efficiency of PCR amplification for both primer pairs is 2 (Livak and Schmittgen, 2001). PCR amplification efficiencies for the primer pairs must be established by means of calibration curves.

 Lines 174-176. It should read “Further analysis revealed 26, 24, 37 and 308 DEPs in HE-â…  versus HE-â…¡, HE-â…¡ versus HE-â…¢, HE-â…¢ versus HE-â…£ and HE-â…£ versus HE-â…¤ respectively” instead of “Further analysis revealed 26, 24 and 37 DEPs in HE-â…  versus HE-â…¡ HE-â…¡ versus HE-â…¢, HE-â…¢ versus HE-â…£ and HE-â…£ versus HE-â…¤ respectively”.

 Lines 252-253. The authors should explain why these proteins (FABP, Nct-1, Shf and Crp) have been chosen.

 Figures 7 and 8. The title of Y-axis must be added.

Author Response

Dear Reviewer,

Thank you very much for your comments and suggestions. The valuable comments from you not only helped us with the improvement of our manuscript, but suggested some ideas for future studies.

Below you will find our responses to your comments:

General comments

  1. Several figures are of low resolution and I could not read the text (Figures 1, 4, 5 and 6).

Response: Thank you for your kindly reminding. Other reviewers also give the same comments. We submit the original images in .pdf format with higher dpi to the journal in revised manuscript.

  1. I think that the following data should be included in the supplementary material in excel file format (these data help readers and reviewers to understand and analyze the results obtained): -The complete lists of differentially expressed proteins DEPs (comparison group, description, fold change and p-value); -The complete list of significantly enriched GO terms and KEGG pathways (description, p-values).

Response: We submit lists of DEPs, enriched GO terms and KEGG pathways in the supplementary material in excel file format in revised submission.

  1. The conclusions are too general. There is no information about the specific results achieved by the authors. I would suggest the authors to rewrite their conclusions.

Response: Thanks for your comments and we rewrite the conclusions as followed.

 “The present study which applied TMT-based proteomics approach in hepatopancreas during female M. nipponense ovarian maturation, provides an overview of regulatory mechanisms of hepatopancreas in crustacean reproduction. The DEPs data in HE-â…  versus HE-Ⅱ,HE-â…¡ versus HE-Ⅲ,HE-â…¢ versus HE-â…£ and HE-â…£ versus HE-â…¤ respectively, indicating the HE-â…¤ has the most different proteins than other four stages. KEGG) enrichment results showed that in hepatopancreas, as the ovaries developed to maturation, carbohydrate metabolism, lipid metabolism, amino acid metabolism and lysosome played important roles. Table 2 and 3 conclude the candidate proteins from this study based on the DEPs and KEGG enrichment analysis. qPCR analysis proves the proteome results was consistent with mRNA expression results. Further investigation of 4 randomly selected candidate proteins (fatty acid-binding protein, NPC intracellular cholesterol transporter 1, Serine hydroxymethyltransferase and Crustapin) showed their involvement in ovary maturation. So far, more efforts have been made in screening key pathways, pro-teins and genes related to ovary maturation of M. nipponense and nutrient metabolism-related pathways and proteins are becoming increasingly prominent. For further study, we will aim to analyze the spatial and temporal expression patterns and biological functions of these candidate proteins and their regulation relationship in ovarian maturation.”

  1. English should be revised because some sentences are not clear.

Response: Thanks for your kind comments. The revised manuscript has been edited and proofread by an editing company in Ireland.

 Specific comments

  1. Line 29. Please, explain, in the abstract, the meaning of the acronyms HE-I, HE-II, HE-III, HE-IV and HE-V.

Response: We add an explanation in the abstract. “HE-â… , HE-â…¡, HE-â…¢, HE-â…£ and HE-â…¤ means hepatopancreas sampled from ovary stageâ…  to â…¤.”

  1. Lines 37-38. It should read “were involved in ovary maturation” instead of “were involved ovary maturation”

Response: Here we changed to “were involved in ovary maturation”.

  1. Lines 61-62. Please, revise this sentence: “ nipponense has characteristic with short sexual maturation cycle”.

Response: Here we corrected to “Adult female M. nipponense have a short sexual maturity cycle.”

  1. Lines 119-120: It should read “About 60 fractions of the eluted peptides were combined in 12 fractions and drained by vacuum concentrator” instead of “About 60 fractions of the eluted peptide were combined 12 fractions and drained by vacuum concentrator”

Response: Here we changed to “About 60 fractions of the eluted peptides were combined in 12 fractions and drained by vacuum concentrator.”

  1. Line 122. FA: formic acid?

Response: Yes, FA means formic acid. We corrected this sentence.

  1. Lines 122-123. Please, revise this sentence: “Each sample was dissolved in solvent A/B (A: 0.1% FA in water; 80% acetonitrile with 0.1% FA), and was centrifuged for 2 min at 20000 g”. Which is the solvent B?

Response: We corrected this sentence to “Each sample was dissolved in solvent A/B (A: 0.1% formic acid in water; B: 80% acetonitrile with 0.1% formic acid), and was centrifuged for 2 min at 20000 g.”

  1. Lines 138-139. Has the P-value been adjusted for multiple testing correction (for example by the false discovery rate).

Response: Yes, The P values were corrected by the false discovery rate (p<0.05).

  1. Lines 151-152. The 2-ΔΔCT method used to analyze the mRNA expression levels is valid only if the efficiency of PCR amplification for both primer pairs is 2 (Livak and Schmittgen, 2001). PCR amplification efficiencies for the primer pairs must be established by means of calibration curves.

Response: We agree with your comments and yes, the amplification efficiency of all primers was tested by establishing standard curve before the qPCR validation was performed (Primers with amplification efficiency above 90% were used for the further step of qPCR). We add the explanation in the ‘2.5 Real-time PCR validation and statistical analysis”.

  1. Lines 174-176. It should read “Further analysis revealed 26, 24, 37 and 308 DEPs in HE-â… versus HE-â…¡, HE-â…¡ versus HE-â…¢, HE-â…¢ versus HE-â…£ and HE-â…£ versus HE-â…¤ respectively” instead of “Further analysis revealed 26, 24 and 37 DEPs in HE-â…  versus HE-â…¡ HE-â…¡ versus HE-â…¢, HE-â…¢ versus HE-â…£ and HE-â…£ versus HE-â…¤ respectively”.

Response: Thanks for your comments. We corrected here to “Further analysis revealed 26, 24, 37 and 308 DEPs in HE-â…  versus HE-â…¡, HE-â…¡ versus HE-â…¢, HE-â…¢ versus HE-â…£ and HE-â…£ versus HE-â…¤ respectively”.

  1. Lines 252-253. The authors should explain why these proteins (FABP, Nct-1, Shf and Crp) have been chosen.

Response: For candidate key proteins screening, first we defined the differentially expressed proteins (DEPs) according to the expression quantitation fold change. Then the DEPs were also subjected to functional GO terms and KEGG pathways enrichment. The top 3 up/down regulated DEPs and the ones which were in GO and KEGG enrichment pathways with P<0.05, were the candidate key proteins. All these candidate key proteins were listed in Table 2 and Table 3. Fifteen DEPs with large expression quantitation fold change in four different comparison groups (HE-â…  vs. HE-â…¡, HE-â…¡ vs. HE-â…¢, HE-â…¢ vs. HE-â…£, HE-â…£ vs. HE-â…¤), were selected from Table 2 and Table 3 for qPCR validation. For further investigation, 4 proteins were selected randomly from 15 validated candidate key proteins, one in each comparison groups. We also corrected the description in the manuscript.

  1. Figures 7 and 8. The title of Y-axis must be added.

Response: We add the title of Y-axis in these two figures.

Reviewer 3 Report

Manuscript titled " Hepatopancreas proteomic analysis reveals key proteins and pathways in regulatory of ovary maturation of Macrobrachium nipponense " by Jiang et al. aims a study of the important role in ovarian maturation in hepatopancreas of the crustacean Oriental river prawn Macrobrachium nipponense.

In this study, authors used a valid methodological approach to indicate the key proteins of carbohydrate metabolism, lipid metabolism, amino acid metabolism and lysosome pathways.

For this purpose, they explored differentially expressed proteins using liquid chromatography-tandem mass spectrometry (LC-MS/MS) proteomics approach and mRNA expression by qPCR analysis.

Study is original and fairly well structured. Approach, methodologies and analysis used in this work are satisfactory. Results are well presented in a comprehensive way and sufficiently discussed. Its content in general justifies the length. Language is clear and understandable.

In my opinion, this manuscript deserves to be published in Animals because it fits with the aim of the Journal. In fact, it provides comprehensive analytical results and it offers new insight into regulatory mechanisms of hepatopancreas in crustacean reproduction.

I believe this work is a good reference for other researchers.

Minor comments

Although most of the figures have good quality and well-illustrated, Figures 4, 5 and 6 showed a low resolution denying the correct reading of the text and symbol legends. I recommend the submission of higher dpi figures.

Table S1, Singular of Primers is “Primer”. Moreover, I suggest to change F with Forward and R with Reverse. Please, correct in the header

L296-298 This sentence is not clear and it seems not finished. Please, change it and put with a clear meaning

In References L451, L470, L487, L585, please revise the format of authors, year, name of journal.

Author Response

Dear Reviewer,

Thank you very much for your comments and suggestions. The valuable comments from you not only helped us with the improvement of our manuscript, but suggested some ideas for future studies.

Below you will find our responses to your comments:

Minor comments

  1. Although most of the figures have good quality and well-illustrated, Figures 4, 5 and 6 showed a low resolution denying the correct reading of the text and symbol legends. I recommend the submission of higher dpi figures.

Response: Thanks for your kind suggestions and we submit the original images in .pdf format with higher dpi to the journal in revised manuscript.

  1. Table S1, Singular of Primers is “Primer”. Moreover, I suggest to change F with Forward and R with Reverse. Please, correct in the header.

Response: Thanks for your comments and we correct these in the revised Table S1. 

  1. L296-298 This sentence is not clear and it seems not finished. Please, change it and put with a clear meaning

Response: Thanks for your comments and we rewrite this part to make it more clearly. “Hepatopancreas transcriptomes of 5 ovary stages were compared to gain new insights into the role of the hepatopancreas in ovarian maturation in M. nipponense, changes in proteins identified in hepatopancreas will provide further information for function investigation of the molecular regulatory mechanisms in crustaceans [22].”

  1. In References L451, L470, L487, L585, please revise the format of authors, year, name of journal.

Response: Thanks for your comments and we correct these errors in References.

Round 2

Reviewer 1 Report

I have no more questions.

Author Response

Thanks very much for good suggestions and comments.

Reviewer 2 Report

The manuscript has been improved and most of my previous comments were adequately addressed.

Specific comments

Please, revise the sentences listed below (these sentences are unclear or have errors):

Lines 77-78. The study aimed s to detect potential functional proteins and signaling pathways in M. nipponense hepatopancreas involved in regulating ovarian maturation

 Lines 78-80. The results will provide new insight into the regulatory mechanisms involved in the role of the hepatopancreas in crustacean reproduction of.

Lines 86-87. The different stages of varian development were determined based on color according to previous study

Lines 152-153. The gene encoding eukaryotic translation initiation factor 5A. EIF was used as a reference gene

 Lines 155-156. T Tissues (H: heart, G: gill, HE: hepatopancreas, O: ovary, M: muscle) were used for total RNA extraction.

Lines 167-168. In total, 17,999 peptides were detected in the M. nipponense hepatopancreas proteome. Fig. 2A–E.

Lines 314-316. Based on the  current results, there were relatively few DEPs between the HE-â…  versus HE-â…¡, HE-â…¡ versus HE- â…¢, HE-â…¢ versus HE- were rarely.

Lines 343-344. Yolk and lipid production began at ovary stage â…¡ s reflected by the gradual enlargement of the ovary and the color gradually turning yellow.

Lines 355-358. In this study, the mRNA expression of Nct-1 encodes positively correlated with ovarian maturation in hepatopancreas but negatively correlated with ovarian development, suggesting that it was involved in steroid hormone synthesis that inhibited ovarian maturation.

Lines 372-374. Previous research showed that glucose, pyruvate, glutamine and glycine metabolism increase significantly during maturation of boving oocytes [53]. However, the mechanisms involved in oocytes maturation squire furtherinvestigation.

Lines 383-386. The comparison between these two stages also confirmed this phenomenon as indicated by the numerous. DEPs in a large number of differentially expressed proteins and signaling pathways.

Lines 391-392. Aspartate aminotransferase controls enzyme activity during the spawning phase in fish as a the central metabolic [59].

Line 403. Protein lgg-1 were one of important member in autophagy.

Author Response

Dear Reviewer,

Thank you very much for your comments. Below you will find our responses to your comments:

Specific comments

Please, revise the sentences listed below (these sentences are unclear or have errors):

  1. Lines 77-78. The study aimed s to detect potential functional proteins and signaling pathways in M. nipponense hepatopancreas involved in regulating ovarian maturation

Response: Here we corrected to “The study aimed to detect potential functional proteins and signaling pathways in M. nipponense hepatopancreas involved in regulating ovarian maturation.”

  1. Lines 78-80. The results will provide new insight into the regulatory mechanisms involved in the role of the hepatopancreas in crustacean reproduction of.

Response: Here we corrected to “The results will provide new insight into the regulatory mechanisms involved in the role of the hepatopancreas in crustacean reproduction.”

  1. Lines 86-87. The different stages of varian development were determined based on color according to previous study

Response: Here we corrected to “The different stages of ovarian development were determined based on color according to previous study.”

  1. Lines 152-153. The gene encoding eukaryotic translation initiation factor 5A. EIF was used as a reference gene

Response: Here we changed to “The eukaryotic translation initiation factor 5A gene (EIF) was used as a reference gene.”

  1. Lines 155-156. T Tissues (H: heart, G: gill, HE: hepatopancreas, O: ovary, M: muscle) were used for total RNA extraction.

Response: Here we corrected to “Tissues (H: heart, G: gill, HE: hepatopancreas, O: ovary, M: muscle) were used for total RNA extraction.”

  1. Lines 167-168. In total, 17,999 peptides were detected in the M. nipponense hepatopancreas proteome. Fig. 2A–E.

Response: Here we deleted “Fig. 2A–E.”

  1. Lines 314-316. Based on the current results, there were relatively few DEPs between the HE-â… versus HE-â…¡, HE-â…¡ versus HE- â…¢, HE-â…¢ versus HE- were rarely.

Response: Here we corrected to “Based on the current results, there were relatively few DEPs between the HE-â…  versus HE-â…¡, HE-â…¡ versus HE- â…¢, HE-â…¢ versus HE- â…£.”

  1. Lines 343-344. Yolk and lipid production began at ovary stage â…¡ s reflected by the gradual enlargement of the ovary and the color gradually turning yellow.

Response: Here we corrected to “Yolk and lipid production began at ovary stage â…¡ reflected by the gradual enlargement of the ovary and the color gradually turning yellow.”

  1. Lines 355-358. In this study, the mRNA expression of Nct-1 encodes positively correlated with ovarian maturation in hepatopancreas but negatively correlated with ovarian development, suggesting that it was involved in steroid hormone synthesis that inhibited ovarian maturation.

Response: Here we changed to “In this study, the mRNA expression of Nct-1 was positively correlated with ovarian maturation in hepatopancreas while negatively in ovary, suggesting that it was involved in steroid hormone synthesis that inhibited ovarian maturation.”

  1. Lines 372-374. Previous research showed that glucose, pyruvate, glutamine and glycine metabolism increase significantly during maturation of boving oocytes [53]. However, the mechanisms involved in oocytes maturation squire furtherinvestigation.

Response: Here we corrected to “Previous research showed that glucose, pyruvate, glutamine and glycine increased significantly during maturation of boving oocytes [53]. However, their mechanisms involved in oocytes maturation required further investigation.”

  1. Lines 383-386. The comparison between these two stages also confirmed this phenomenon as indicated by the numerous. DEPs in a large number of differentially expressed proteins and signal-ing pathways.

Response: Here we corrected to “The comparison between these two stages also confirmed this phenomenon as indicated by the numerous DEPs and signaling pathways.”

  1. Lines 391-392. Aspartate aminotransferase controls enzyme activity during the spawning phase in fish as a the central metabolic [59].

Response: Here we corrected to “Aspartate aminotransferase controls enzyme activity during the spawning phase in fish central metabolism [59].”

  1. Line 403. Protein lgg-1 were one of important member in autophagy.

Response: Here we corrected to “Protein lgg-1 was one of important members in autophagy.”
